# Organic Ingredients as Alternative Protein Sources in the Diet of Juvenile Organic Seabass (*Dicentrarchus labrax*)

**DOI:** 10.3390/ani13243816

**Published:** 2023-12-11

**Authors:** Eslam Tefal, Ignacio Jauralde, Silvia Martínez-Llorens, Ana Tomás-Vidal, María Consolación Milián-Sorribes, Francisco Javier Moyano, David S. Peñaranda, Miguel Jover-Cerdá

**Affiliations:** 1Aquaculture and Biodiversity Research Group, Institute of Science and Animal Technology (ICTA), Universitat Politècnica de València, 46022 Valencia, Spain; igjaugar@doctor.upv.es (I.J.); silmarll@dca.upv.es (S.M.-L.); atomasv@dca.upv.es (A.T.-V.); mamisor@etsia.upv.es (M.C.M.-S.); dasncpea@upv.es (D.S.P.); mjover@dca.upv.es (M.J.-C.); 2Department of Animal and Poultry Production, Faculty of Agriculture, Damanhour University, Damanhour 22516, Egypt; 3Departamento de Biología y Geología, Facultad de Ciencias, Campus de Excelencia Internacional del Mar (CEI-MAR), Universidad de Almería, 04120 Almeria, Spain; fjmoyano@ual.es

**Keywords:** organic aquaculture, alternative protein sources, organic ingredients, seabass, fish nutrition, aquaculture sustainability

## Abstract

**Simple Summary:**

This study explores the use of organic ingredients as protein sources in the diet of juvenile organic seabass. Various diets, including organic options like insect meal, Iberian pig byproduct, and rainbow trout meal byproduct, were compared to a control diet using conventional fishmeal. The research found that the control diet produced the best growth rates, and histological analysis indicated certain differences. While organic ingredients promise to replace fishmeal in aquaculture diets, further investigation is needed to achieve a complete substitution.

**Abstract:**

The use of organic ingredients as a source of protein in aquaculture diets has gained significant attention due to the growing demand for organic seafood products. This study aimed to evaluate the potential for the use of organic ingredients as protein sources in the diet of juvenile organic seabass (*Dicentrarchus labrax*). A total of 486 juvenile seabass with an average weight of 90 g were fed six diets containing varied organic proteins. The control group (CON) was fed a diet with conventional fishmeal from sustainable fisheries as the primary protein source. The other five groups were fed diets with different compositions: organic Iberian pig meal byproduct (IB diet), a combination of organic Iberian pig meal byproduct and insect meal (IB-IN diet), a mix of organic Iberian pig meal byproduct and organic rainbow trout meal byproduct (IB-TR diet), a blend of organic rainbow trout meal byproduct and insect meal (TR-IN), and a mixed diet containing all of these protein sources (MIX diet). Over a 125-day feeding trial, growth performance, feed utilisation, feed digestibility, and histological parameters were assessed. The results showed that the fish fed the control diet had the highest final weight and specific growth rate, followed by the fish fed the TR-IN and IB-TR diets. The IB-TR diet had the highest apparent digestibility coefficients (ADCs) for protein, while the TR-IN diet had the lowest. Histological analysis revealed that fish fed the control diet had the largest nucleus diameter and hepatocyte diameter. Use of IN seems to penalise performance in several ways. Fish fed diets containing insect meal grew less, and those diets had lower digestibility. Fish fed the TR and IB diets grew at rates near that of the control, and the feed had acceptable digestibility.

## 1. Introduction

The term “organic production” refers to a farming and food-production approach that incorporates optimal environmental practices, promotes extensive biodiversity, conserves natural resources, upholds stringent standards for animal welfare, and aligns with the preferences of consumers seeking products created using environmentally friendly methods [1]. Organic aquaculture is a modest but growing part of the global food-supply chain [2]. Its production methods [3] were adopted due to the increasing interest in sustainable resource utilisation [4,5]. Organic aquaculture can be carried out using various technologies, including recirculating aquaculture systems (RAS), net pens, cages, raceways, and tanks [6]. The appropriate fish stocking density in RAS qualifies these systems for use in organic production, providing safe conditions for most fish in terms of animal welfare and biosecurity concerns [7]. Nevertheless, transitioning from conventional to organic aquaculture involves an intricate and multifaceted process, encompassing considerations related to consumer safety, ecological and environmental impacts, socioeconomic factors, and animal welfare [8,9]. The argument over the use of organic feeds for organic aquaculture is still ongoing because a balance must be struck between the realities of the supply of sources for aquafeeds and the fundamental principles of organic food production. Additionally, feeds must support animal health and growth, provide a final edible product of excellent quality, have a low impact on the environment, and be balanced to meet the nutritional needs of the farmed species [10].

In 2020, the EU27’s total organic aquaculture production was approximated at 74,032 tonnes, constituting 6.4% of the EU’s total output. This production marks a 60% increase from 2015 (46,341 tonnes at the EU 27 level in 2015), primarily due to increased production of organic mussels [11]. One of the main species produced organically in the Mediterranean is European seabass, the production of which increased from 2000 tonnes in 2015 to 2750 tonnes in 2020. Greece is the leading EU producer of this fish [11]. Economic considerations such as increased production expenses and increased retail costs have dissuaded both farmers and consumers, limiting the growth of organic production of seabass [12]. Research on consumer preferences shows that organic seabass production in the Mediterranean is economically promising [13,14], but appropriate marketing strategies still need to be developed [15,16]. Furthermore, a major obstacle to expanding organic aquaculture production is the lack of organic feeds, particularly for carnivorous species. The limitations imposed by the EU organic regulations make it difficult to find organic feed ingredients that are rich in protein and thus to design well-balanced organic diets [4,17,18,19].

More research is required on the organic cultivation of seabass. Previous studies reported that organic fish show improved growth performance, lower feed-conversion ratios, and increased metabolic rates compared with fish grown in conventional aquaculture [18]. However, other studies have found no differences in stress and immunological indices between fish grown in organic and conventional aquaculture [18,20].

The general principles of ecological production, such as the development of processes that are based on environmental systems and that use the system’s natural resources, the restricted use of synthetic substances, and the limited use or non-use of genetically modified organisms (GMOs), apply to organic aquaculture, with some additional limitations regarding the availability of organic resources [21]. The relevant regulatory limitations stipulate a maximum of 60% organic plant ingredients and the absence of synthetic amino acids [22]. Continual efforts are required to identify alternative sources of nutritional protein and lipids for organic feeds in organic aquaculture, with a focus on minimising the utilisation of fishmeal (FM) and fish oil in such feeds. However, there is a need to focus on the quality and certification of alternative ingredients for use in organic aquaculture. Research is still being done to investigate novel alternative formulations of ingredients and the quality of the resulting products [1].

Many researchers have studied the effects of substituting FM with plant-based proteins [23,24,25,26,27,28]. Completely replacing animal proteins with plant proteins has generally not been successful due to concerns about antinutrients, changes in amino-acid absorption, potential micronutrient deficiencies, and immune suppression [29,30,31]. Other potential feed sources, excluding plant proteins, include microbiological organisms (bacteria, microalgae, fungi), byproducts from terrestrial animals (processed animal protein (PAP), blood meal), annelid worms obtained from wild harvesting and cultivation, and the larvae and pupae of insects [32,33,34]. The utilisation of animal byproducts is made possible by PAP, a key ingredient in feeds [35]. According to several studies, insects can be used as a source of protein for fish [36,37,38,39]. European seabass can be fed insect meal from *Tenebrio molitor* at varying concentrations without adversely affecting growth performance, according to a feeding assay [40]. Byproduct meals can be highly appealing due to their competitive pricing compared to fish meal, making them a potentially interesting and cost-effective option [41].

Due to regulatory restrictions, it is difficult to find enough organic protein sources suitable for seabass, one of the main carnivorous fish produced in Europe. Transformed animal proteins (TAPs) from non-ruminant animals, whose use is permitted in conventional aquaculture (RD 578/2014), as well as insects (Regulation EU 893/2017), are suggested. The use of organic-derived TAPs in organic aquaculture does not violate any regulations and facilitates the formulation of organic aquaculture feed without the need for captured FM, relying solely on the recovery of byproducts from organic aquaculture.

On the other hand, there has been considerable interest in the use of in vitro assays to evaluate the digestibility of a prospective feed product for aquatic species, such as fish, prawns, and molluscs [42]. The in vitro digestibility test is appropriate for preliminary research. It allows many samples to be analysed because it is inexpensive, has no ethical restrictions, and is reasonably simple to carry out [43]. It also allows the conduction of controlled experiments to investigate how proteins, lipids, and carbohydrates in feed items are hydrolysed [42]. Research on fish digestion in vitro is still in its infancy, based on the number of relevant publications.

The present work aimed to provide a 100% organic diet for seabass, one of Europe’s most important marine aquaculture species, using alternative organic raw materials such as insects, byproducts from Iberian pigs, and rainbow trout remains. This research may support a dramatic improvement in organic aquaculture.

## 2. Materials and Methods

### 2.1. System for Rearing

The trial was conducted in 18 cylindrical fiberglass tanks, each with a volume of 1750 L, as part of a saltwater recirculating system with a total capacity of 75 m^3^. The system was equipped with a rotating mechanical filter, a gravity biofilter with a capacity of 6 m^3^, and a skim (September to January). A heat pump was used to ensure that the water temperature remained constant (20.9 °C), and all tanks had aeration. The dissolved oxygen level was 7.7 mg L^−1^, and salinity was 31.3 g L^−1^. The pH was maintained at 8.0, with nitrates (NO^−3^) at a concentration of 33.2 mg L^−1^, nitrites (NO^−2^) at 0.13 mg L^−1^, and ammonium (NH^+4^) at 0.03 mg L^−1^. The photoperiod was natural (11 h), and the lighting was the same in all tanks.

### 2.2. Fish

Juvenile organic seabass from the fish farm Sonrionansa S.L. situated in Pesues (Cantabria, Spain) were delivered to the Universitat Politècnica de València and distributed among experimental tanks. Before the feeding experiment, a 15-day acclimatisation period was provided to allow all fish to adapt to the laboratory conditions. There were 486 fish, with an average weight of 90 g, distributed throughout the 18 test tanks (27 fish per tank). The experiment was carried out over 125 days.

### 2.3. Diets and Feeding

The proximal composition of the raw materials is shown in Table 1. Six diets were tested in triplicate: (1) a control diet containing FM provided for sustainable fisheries as a protein source (CON); (2) a diet in which the protein source was composed of Iberian pig meal byproduct (diet IB); (3) a diet containing organic Iberian pig meal byproduct and organic insect meal (diet IB-IN); (4) a diet based on organic Iberian pig and organic rainbow-trout byproduct meal (diet IB-TR); (5) a diet with organic rainbow trout byproducts and organic insect meal as protein sources (diet TR-IN); and (6) a MIX diet containing organic insect meal, organic rainbow trout meal and organic Iberian pig byproduct meal (Table 2). The nutritional compositions of all the diets are represented as the means of five separate analyses conducted during each feed-manufacturing cycle. Formulations were initially designed with different amounts of raw ingredients to maintain a consistent composition of 45% crude protein (CP) and 20% crude lipid (CL); challenges in effectively mixing certain ingredients led to variations in some of these values (Table 2).

All diets were produced at the Universitat Politècnica de València using a semi-industrial twin-screw extruder (CLEXTRAL BC-45, Firminy, St Etienne, France), under the following processing conditions: a screw speed set at 100 rpm, pressure ranging from 40 to 50 atm, and a temperature maintained at 110 °C. Calcium phosphate and organic vegetable amino acids were included in diets as supplements (lysine and methionine). Diet formulation and manufacture were carried out with organic raw materials labelled and approved by Regulation (EU) 2018/848.

Each experimental diet was evaluated in three randomly distributed tanks. The fish were manually fed twice a day at 9:00 and 17:00, six days per week (from Monday to Saturday).

The fish were fed until they reached satiety, and the pellets were administered gradually. Daily monitoring and weighing occurred every 30 days before anaesthetising the fish were anaesthetised with clove oil, which contained 87% eugenol (Guinama^®^, Valencia, Spain), at a concentration of 10 mg L^−1^ of water.

The aim of this process was to evaluate the growth of the fish throughout the experiment, define growth parameters, and assess the overall health of the fish. The fish were starved the day before they were weighed. Ten fish were collected at the start of the experiment and preserved at −30 °C for subsequent analysis of their body composition. Three specimens from each tank were randomly selected for sampling and pooling to determine the approximate composition and amino acid content of their bodies.

### 2.4. Analysis of Nutritional Composition and Amino Acids

The diets and their approximate composition (Table 2), as well as the whole fish, were examined using the methods described in [44] and analysed for the following metrics: dry matter (105 °C to constant weight); ash (incinerated at 550 °C for five hours); crude protein (determined by the direct combustion method DUMAS using LECO CN628, Geleen, The Netherlands); and crude lipid (extracted with methyl-ether using ANKOMXT10 Extractor (Macedon, NY, USA)). Each analysis was carried out in triplicate.

A Waters HPLC system (Waters 474, Waters, Milford, MA, USA) composed of two pumps (Model 515, Waters), an autosampler (Model 717, Waters), a fluorescence detector (Model 474, Waters), and a temperature-control module was used to analyse the levels of amino acids (AA) in the diets and in the fish using the procedure previously described by Bosch et al., 2006 [45].

Before hydrolyzation, aminobutyric acid was introduced as a internal standard. AQC was used to derivatise AA (6-aminoquinolyl-N-hydroxysuccinimidyl carbamate). After oxidation with performic acid, methionine and cysteine were identified individually as methionine sulphone and cystic acid. AA was converted to methionine and cystine after it was separated with a reverse-phase C-18 column by Waters Acc—Tag (150 mm 3.9 mm). Table 3 shows the essential amino acids (EAA) content of the experimental diets. All amino acid analyses were carried out in duplicate.

### 2.5. Indices of Growth

At the end of the trial, the nutrient efficiency indices and growth were determined. Metrics included the survival rate (SR), specific growth rate (SGR), feed intake (FI), feed conversion ratio (FCR), and protein efficiency ratio (PER), considering each tank as an experimental unit. All fish were weighed. Additionally, the productive protein value (PPV%) and productive fat value (PFV%) were calculated. These parameters were calculated using the following equations:SGR = 100 × ln (final weight/initial weight)/days
FI (g 100 g fish − 1 day − 1) = 100 × feed consumption (g)/average biomass (g) × days
FCR = feed consumption (g)/biomass gain (g)
PER = biomass gain (g)/protein intake (g)
PPV% = Protein retained (final fish protein × Final biomass (g)) × 100 − Initial fish protein × initial biomass (g)/Protein ingested (kg of ingested feed × % crude protein)
PFV% = Fat retained (final fish fat × Final biomass (g)) × 100 − Initial fish fat × initial biomass (g)/fat ingested (kg of feed × % crude fat)

### 2.6. Digestibility Assay

The digestibility test was performed after the growth experiment ended and was carried out in triplicate, in three tanks. Fifteen seabass were randomly placed in each experimental tank (190 L fibreglass tank, 88 cm high, 62 cm wide, and 188 cm deep) in a semi-closed recirculating system based on the Guelph system (the faecal material being collected in a settling column). The water flow velocity was altered to reduce the settling of faeces in the drainpipe and increase the recovery of faeces in the settling column.

The fish received one meal per day at 10:00 a.m. The diet was offered so as to reduce waste while the fish were actively feeding. The drainpipe and the settling column were dusted an hour after feeding to prevent faeces from being contaminated by column diets. The faeces were gravity-collected in a plastic container from the base of the settling column 6–7 h after feeding.

After collection, the faeces were weighed and dried in a 60 °C oven for 48 h before analysis. Subsequently, they were preserved in sealed plastic containers and analysed for nutritional components and inert markers. Chromic oxide (Cr_2_O_3_) was used (5 g kg^−1^) as an inert and indigestible marker. An atomic absorption spectrometer was used to determine the amount of chromium oxide in diets and faeces after acid digestion (Perkin Elmer 3300, Perkin Elmer, Boston, MA, USA). Additionally, analyses were conducted for crude protein, dry matter, calcium, energy, and phosphorus in diets and faces. All analyses were performed in triplicate.

The apparent digestibility coefficients of the diet (ADC) were determined using [46]. The ADCs of the dry matter (ADCdm, %) of the diets were determined per Equation (1):ADCdm % = 1 − (% Cr_2_O_3_ in diet/% Cr_2_O_3_ in faeces)(1)

The percentage of ADCs for each dietary nutrient (protein, energy, calcium, and phosphorus) was calculated using Equation (2):ADCnut = 1 − ((marker diet/marker faeces × (nutrient faeces/nutrient diet))(2)

The variables “nutrient diet (g kg^−1^)” and “nutrient faeces (g kg^−1^)” in this equation indicate the amounts of a nutrient (such as protein or energy) in the diet and the faeces, respectively. The measurements “marker diet” (g kg^−1^) and “marker faeces” (g kg^−1^) indicate the amount of marker in the diet and the faeces, respectively.

### 2.7. In Vitro Hydrolysis Assay

The in vitro hydrolysis trial was conducted under conditions that simulated the digestive system of juvenile European seabass [47]). Ten juvenile seabasses with an average weight of 100 g were used and sampled six hours after feeding to ensure the presence of enzymes in both the stomach and the intestine. The fish were euthanised in ice-cold water with a small quantity of clove oil, which acted as an anaesthetic. Subsequently, the fish were immediately dissected to extract the digestive tract. The digestive tract was divided into two parts: (1) the proximal intestine, which encompassed the diffuse pancreas and pyloric caecum, and (2) the stomach. These tissues were utilised to create extracts for measuring protease activity. The methods used were as follows: acid protease was measured by tyrosine release from haemoglobin hydrolysis at pH 2.5 [48]; alkaline protease was measured by tyrosine release from casein at pH 8.5 [49]; and amylase was measured during the preliminary evaluation of the enzymes of the juvenile European seabass at pH 7.5 [50]. The extracts were prepared by mechanical homogenisation of the tissues in distilled water (1:10 *w*/*v*) and centrifugation (3220× *g*, 20 min, 4 °C). The supernatant was then filtered through a dialysis system with a MWCO of 10 kDa (Pellicon XL, Millipore, Burlington, MA, USA), and the concentrated extracts were freeze-dried until they were required for the assays. The activities of acid protease in the stomach (pepsin) and total intestinal alkaline proteases present in the extracts were measured using the methods described in refs. [47,48]. Protease activity levels were used as indicators to estimate the amount of extracts required to provide physiological enzyme-substrate ratios in the assays. These ratios were calculated considering, on one hand, the average total production of enzyme measured in several fish in relation to their live weight, and on the other, the average intake per meal of fish of such a size, a value obtained from commercial ration tables.

Based on these findings, the average enzyme production was estimated as follows: acid protease, 37.7 U g^−1^ weight; and alkaline protease, 24.7 U g^−1^ weight. The conditions are given in Table 4.

Three assay replicates were conducted for each diet, and a blank sample was also included. The blank sample was created by deactivating the enzymatic extracts through heat treatment before they were added to the bioreactors. This step enabled measurement of the amino acids in the extracts and the diet.

### 2.8. Histological Analysis of the Liver

The liver was collected from was three fish per tank after the growth experiment ended. Samples were preserved in phosphate-buffered formalin (4%, pH 7.4). According to typical histological procedures, all the formalin-fixed tissues underwent regular dehydration in ethanol, were conditioned in ultra-clean environments, and were embedded in paraffin. Transverse sections from each paraffin block were taken using a Shandon Hypercut microtome, then stained for haematoxylin and eosin analysis.

One hundred sections of the liver were examined using an Eclipse E400 Nikon light microscope from Izasa S.A. in Barcelona, Spain. To determine the effects of different feeds on the liver, the diameters of hepatocytes and nuclei were measured [51,52].

### 2.9. Statistical Analysis

All data were checked for normal distribution and homogeneity of variances. Using the Statgraphics^®^ Plus 5.1 statistical programme (Statistical Graphics Corp, Rockville, MO, USA), various growth and nutrient indices, retention of AA, ADC, in vitro hydrolysis, and histological measurements were analysed using analysis of variance with a Newman-Keul test for multiple comparisons. The initial covariate weight was used to analyse growth indices. The findings are represented as means with standard error (SEM, standard error of the mean). The significance level was established at *p* < 0.05.

## 3. Results

### 3.1. Growth and Nutritional Parameters

Fish weights increased throughout the experiment regardless of the experimental group. Figure 1 illustrates that the control group had the highest final weight, followed by the fish diets containing organic fishmeal (TR-IN and IB-TR), while the fish in the remaining treatment groups weighed less.

The final weight of the fish (FW) and the SGR were affected by the composition of the diet (Table 5). Fish fed the control diet had the highest final weight and SGR (258.7 g, 1.16%/day, respectively), followed by fish fed diets TR-IN and IB-TR, which obtained grew more than fish fed the IB-IN diet. Survival, FI, and FCR did not show significant differences at the end of the experiment.

### 3.2. Body Composition, Nutrient Retention Efficiency

The nutritional composition of the whole body and the retention efficiency of protein and fat are shown in Table 6. No significant differences were observed for total dry matter, protein, fat, ash, protein efficiency retention, or fat efficiency retention (PPV and PFV). Table 6 also shows the efficiency of EAA retention. Except for arginine (Arg) and histidine (His), there were no statistical differences in the retention efficiency of EAA between the experimental diets. The highest retention for Arg was shown in fish fed the TR-IN diet, and the highest retention for His was observed in fish fed the TR-IN and IB-TR diets. Generally, the lowest EAA retention efficiency was found in fish fed the IB and MIX diets.

### 3.3. Digestibility

The ADC for dry matter, phosphorus, protein, and energy of the experimental diets are shown in Table 7. ADCs for dry matter and phosphorus fish fed the IB diet showed the highest values (62.2 and 68.3%, respectively), and the TR-IN diet was the lowest (43.3 and 25.0%, respectively). The IB-TR diet was associated with the highest values (91.8%) of protein ADCs, and the TR-IN diet gave the lowest values (85.3%). A fish fed the IB diet showed the highest-energy ADCs.

### 3.4. In Vitro Hydrolysis Assay

Results for amino acids released through the membrane after the dietary protein hydrolyzation sampling point are shown in Figure 2. The hydrolysis values were very similar among diets.

The results were used to adjust the protein-hydrolysis dynamics in each diet. Using the following equations, various hydrolysis rates were estimated for a specific amount of protein. For example, for 100 mg of protein in the diet, the total hydrolysis time ranges from 9 to 12 h, as presented in Table 8. Furthermore, the steeper slope of the IB diet’s adjustment line suggests that it hydrolyses faster. Table 8 shows the linear equations representing the relationships between the degree of protein hydrolysis (y) and the time (x) for each diet, along with the corresponding time points.

Based on the hydrolysis rate, the IB-TR, TR-IN, and IB diets required less time for protein hydrolysis than did the rest of the diets, especially the CON diet, which may have important implications when digestion transit rates and feeding frequencies are considered.

### 3.5. Histology of the Liver and Intestinal

The liver histology results for seabass fed experimental diets is shown in Table 9 and shows differences in hepatocyte measurement (nucleus and diameter of the hepatocyte). Fish fed the CON diet showed the largest nucleus diameter, followed by fish fed IB-TR and TR-IN, then fish fed IB-IN, IB, and MIX diets, with smaller nucleus diameters. Likewise, fish fed the CON diet exhibited larger hepatocyte diameters fish fed the experimental diets. No differences were found among the fish fed experimental diets.

The liver histology of seabass fed experimental diets is shown in Figure 3. The histology of the liver for each treatment revealed hepatocytes with irregular morphology, absence of necrosis, and large nuclei displaced from the centre to peripheral areas. Lipid accumulations forming vacuoles in the hepatocyte cytoplasm were observed mainly in fish fed the CON, IB, and IB-IN diets.

## 4. Discussion

Formulating organic diets for carnivorous fish is a challenging task that requires careful consideration of fish nutritional requirements, organic regulations, and the availability and quality of organic feed ingredients. Perfectly balancing the amino acid profile is not always possible because organic feeding standards prohibit the use of synthetic amino acids. The organic amino acids available are precursors of amino acids and amino acid mixtures, making it impossible to achieve the perfect balance, as can be done with synthetic amino acids. The performance of fish fed organic alternative ingredients must be studied to optimise organic feeding practices. Improving the sustainability of carnivorous-fish production is one of the primary objectives of replacing fishmeal (FM), along with FM’s high cost, which makes more affordable alternatives attractive.

### 4.1. Fish Performance

The results of this research suggest that organic ingredients, such as insects, byproducts of Iberian pigs, and rainbow trout remains, have potential as part of an initial step in completely replacing fishmeal in European seabass diets in aquaculture. The utilisation of byproducts results in highly competitive prices compared to fish meal, which may incentivise their use even when they produce suboptimal growth [41].

The findings of previous studies comparing conventional and organic feed in European seabass provide valuable insights. However, caution should be exercised when extrapolating those results for comparison to the present experiment due to differences in feed composition. For example, the study by Di Marco et al. (2017) [18] compared feeds containing different concentrations of fishmeal; the organic feed contained 56% fishmeal, and the conventional feed contained only 20%. Similarly, in experiments conducted with seabream, Mente et al. (2012) [5] observed improved growth in fish fed organic feed compared to those fed conventional feed. Again, however, the organic feed contained more fishmeal (63%) than the conventional feed (50%). These findings further support the notion that organic feed formulations enriched with fishmeal can positively impact growth performance in marine species. However, the higher growth seen in organic-fed fish in those experiments cannot be attributed solely to the origin, organic or conventional, of the fishmeal because the concentration is a significant factor. Fishmeal is known to be a protein source of high quality that includes essential amino acids and other nutrients necessary for optimal growth in marine species. Therefore, the higher percentage of fishmeal in the organic feed likely contributed to the observed growth improvement.

The outcomes of the present experiment indicate that the IB-IN, IB, and MIX diets resulted in lower final fish weights, suggesting a negative impact on the growth performance of European seabass. These results align with observations from a previous trial conducted by Tefal et al. (2023) [41], which investigated the total substitution of fishmeal (FM) with a new organic raw material in gilthead seabream feeding. Tefal et al. (2023) [41] found that organic diets can replace fishmeal in gilthead seabream feed without negatively impacting growth performance. Nevertheless, the CON diet resulted in the highest final weight. The growth and composition of fish can be significantly influenced by the specific ingredients used in organic diets, such as trout remains, Iberian pig viscera, and insects. In the present experiment with European seabass, the lower final fish weights observed in fish fed the IB-IN, IB, and MIX diets could be ascribed to several factors. Firstly, the diet’s specific composition, including the inclusion levels and quality of alternative protein sources, may have influenced the growth performance of the fish. It is possible that the proportions or sources of alternative proteins used in these diets were not optimal to meet the nutritional requirements of European seabass. Although the diets were formulated to cover the needs for amino acids, differences in amino acid retention could partially explain these results (Table 5). The amino acid composition of feed, particularly the differential levels of essential amino acids such as methionine and lysine, plays a pivotal role in influencing growth performance. The variations in the levels of these organic amino acids can significantly impact the overall nutritional quality of the diet and subsequently affect the growth outcomes observed in our study. Methionine and lysine are essential amino acids crucial for protein synthesis and various metabolic processes in aquatic organisms. The fact that these amino acids were not present at equal levels in all feeds in our study indicates a potential imbalance in the amino acid profile of the diets. This imbalance can lead to limitations in protein synthesis, affecting the overall growth performance of the aquatic organisms.

Additionally, the observed differences in energy levels may have further contributed to the variations in growth outcomes. Energy is a fundamental factor influencing metabolic processes, and disparities in energy content can affect nutrient utilisation and growth efficiency. The disparities in the levels of essential amino acids, specifically methionine and lysine, coupled with differences in energy levels, likely contributed to the observed variations in growth performance.

Other factors, such as feed palatability, digestibility, and overall diet formulation, including essential nutrients and amino acids, can also play a role in growth performance. In previous studies, the use of insect meal as a partial substitute for fishmeal has shown no negative impact on fish performance in terms of growth, feed utilisation, and digestibility [53,54,55,56,57,58]. However, these findings contrast with the results of the present study, wherein the IB-IN diet resulted in poorer growth compared to the other experimental diets. This growth deficiency can be attributed to the low digestibility of the IB-IN diet, which may result in a lack of essential amino acids and reduced nutrient availability. In contrast, the TR-IN and IB-TR diets, which incorporated trout meals, resulted in growth approaching that seen with the control diet; the other experimental diets yielded lower growth. Compared to the other experimental diets, the improved growth observed in European seabass fed the TR-IN and IB-TR diets could be explained by the enhanced amino acid profile and improved digestibility. The inclusion of organic alternative fish ingredients such as rainbow trout remains in European seabass diets may offer cost-effective and environmentally sustainable alternatives to traditional fishmeal or plant-based diets. The lowest SGR was found in seabass fed the IB-IN diet, which may be linked to poor energy availability in the TR-IN diet. IN meals can contain a high percentage of chitin, which is composed of glucosamine and a nitrogen-containing substance found in the exoskeletons of insects [59,60]. This protein has complex effects when it is introduced into fish diets from insect-derived sources. Chitin is commonly recognised for its anti-nutritional characteristics and is generally considered unprocessable by fish [61]. In addition to its detrimental influence on nutrient absorption, chitin has been documented to harm the growth rate and feed conversion of tilapia [62].

### 4.2. Body Composition and Nutrient Retention

The present study revealed no significant differences in body composition among European seabass treatment and control groups, including dry matter, crude protein, crude fat, and ash content. The inclusion of alternative organic ingredients, such as trout remains, Iberian pig viscera, and insects did not significantly impact the body composition of the fish. Similar results have been reported in other studies. Gao et al. (2020) [63] conducted a study on *Cyprinus carpio* fed blood meal and dried porcine soluble and found no notable variances in moisture, crude protein, and ash composition. Another study by Vélez-Calabria et al. (2021) [64] examined the effects of an Iberian pig meal and a vegetable protein blend on seabream. They also reported no significant differences in the protein and ash content of the body among the experimental groups. These findings support the idea that incorporating alternative ingredients into fish diets does not significantly impact body composition in terms of protein and ash content. There is evidence that changes in body composition, especially in body fat, are expected with changes in energy intake. As the intakes in diet were similar, changes in body composition were not expected [65]. These findings showed that incorporating alternative organic ingredients in fish diets, as in the present study, does not result in notable changes in the body composition of European seabass. Nutrient retention and utilisation (PPV and PFV) in the fish were not significantly affected by the inclusion of these ingredients. However, it will be essential to evaluate the long-term effects of these alternative organic diets on body composition and nutrient retention in fish.

The deficit of essential amino acids is a significant challenge when substituting fishmeal with alternative ingredients [66]. The lower retention efficiency of EAA observed in the IB and MIX diets may partially explain the lower final body weights of those fish. This difference can be attributed to imbalances in the amino acid composition of the alternative ingredients. These imbalances can affect the availability and utilisation of essential amino acids, leading to reduced growth and suboptimal muscle efficiency.

The results of the current experiment align with those of previous studies. For example, the TR-IN diet had a similar retention efficiency for Arg (approximately 30%) to that reported by Martínez-Llorens et al. (2012) [67]. The high retention efficiency for Arg and His in fish fed the TR-IN and IB-TR diets agrees with the growth results obtained in the study. The results should be interpreted as indicating that these diets provide a favourable amino acid profile and support improved growth performance in European seabass. Arginine and histidine are essential amino acids that play vital roles in various physiological processes, including protein synthesis, immune function, and antioxidant defence.

### 4.3. Digestibility

Before adding ingredients to a commercial aquaculture feed, it is crucial to determine the nutritional quality of each new protein ingredient. The first stage in this process is to assess apparent digestibility and examine how it affects the growth and welfare of different fish species. The results of this study show that trout meal combined with Iberian pig meal (IB-TR diet) and Iberian pig meal (IB diet) improved protein digestibility in European seabass. The variation in protein ADCs among the diets can be attributed to several factors, e.g., the quality and composition of protein sources. Additionally, the trout and Iberian pig meals yielded higher hydrolysis curves and better histidine retention. It is established that specific amino acids and their ratios in the diet can influence protein digestibility and utilisation by fish [68]. Additionally, most of the organic meals used in the experiment were processed from raw sources, with non-standardised methods used to convert them to meal; the processing methods used to produce trout and Iberian pig meals may have influenced their digestibility.

Another study conducted by Tefal et al. (2023) [41] reported the highest protein ADC value, 93.0%, for the INS diet. However, insect meal is still far from being a standardised product. In fact, in the present study, the diets containing insect meal (TR-IN and IB-IN) yielded lower protein and energy ADCs, a result that can be attributed to antinutrient factors (ANFs). Insect meal is a potential source of protein and other nutrients, but it contains ANFs that can interfere with nutrient digestion and utilisation in fish [60]. The specific ANFs in insect meals vary depending on insect species, rearing conditions, and processing methods. Some common ANFs in insect meals include chitin, protease inhibitors, lectins, and other bioactive compounds. It should be noted that chitin, a major component of insect exoskeletons, is known for its anti-nutritional properties. It is resistant to enzymatic digestion in fish and can impair the breakdown of proteins and the release of energy from the diet [69].

Fish require phosphorus as a mineral element and component of their bones and scales [70]. Fish must use phosphorus effectively because it can affect feed digestibility, farming costs, and water contamination [71]. In the present work, the ADC of phosphorus for European seabass receiving IB diets was higher than that for fish fed other diets. Fish that ingest various types of byproducts in their diets may experience varied levels of phosphorus availability due to changes in size, particle size of bones, density of bones, processing conditions of fishmeal, or the proportion of non-bone to bone phosphorus in fishmeal [72]. The IB diet had the highest dry matter and phosphorus ADCs, proving that it could be an excellent candidate for inclusion in fish feed and even in organic diets due to its good quality. Its inclusion would probably reduce the need for inorganic phosphate in feed. Supporting the excellent quality of IB protein meal, IB alone (IB diet) and IB in combination with TR meal protein (IB-TR diet) yielded excellent results for protein digestibility compared with the other test diets.

### 4.4. In Vitro Hydrolysis Assay

Simulated digestion experiments are incredibly informative and allow the use of fewer live animals while testing novel feed ingredients with species-specific digestive enzymes [73]. Before investing in expensive in vivo animal-feeding tests, these assays can be helpful tools to evaluate the quality of a prospective feed protein [74]. In vitro digestibility techniques were used in the present study to give a preliminary review of the ability of European seabass to hydrolyse proteins in their diet. The diets investigated could be divided into two groups based on the results of the hydrolysis rate: (a) IB-TR and TR-IN and (b) IB, which showed better results than the rest, especially compared to the control. The results of this study are consistent with the result obtained from the digestibility study in vivo, wherein protein ADC was highest in the IB and IB-TR diets. The times required for complete hydrolysis were 9.24, 9.37, 9.89, and 12.22 h, respectively. This result may have significant implications in terms of intestinal transit times and feeding frequencies. Extracts of species-specific enzymes obtained from different sections of the fish digestive system can be used to simulate in vitro digestion [42].

Other fish species, such as *S. aurata*, *S. senegalensis*, and salmonids (mainly rainbow trout and *Oncorhynchus mykiss*), have also been used in the in vitro simulation of fish digestion [42]. The hydrolysis rates obtained with European seabass proteases generally indicate good bioavailability of the proteins in the experimental diet. Similar results were obtained for other ingredients often used in aquatic feeds, such as fishmeal or soybean protein concentrate [75,76]. Other alternative ingredients used in aquafeeds, such as microalgae, have also been found to yield similar results [73]. The existence of alternative organic ingredients in experimental diets with varying rates of protein hydrolysis could have practical relevance in the context of fish nutrition. Highly hydrolysable proteins may rapidly and easily release accessible amino acids, which could stimulate digestion and metabolism. In contrast, the presence of intermediately hydrolysable proteins might lead to a slower rate of amino acid release in the intestinal tract, limiting the saturation of membrane carriers in enterocyte microvilli with amino acids. Importantly, the efficiency of amino acid absorption is influenced by both the relative and total amounts of different amino acids in the intestine.

Discrepancies are frequently observed when comparing crude protein digestibility data obtained through in vivo methods and growth performance. In in vitro hydrolysis assays, as for observations in other animal models, only the bioaccessibility of the protein fraction of the substrate is measured (indicating the susceptibility of this nutrient to enzymatic action). This assessment provides an estimation of potential bioavailability, representing the quantity of amino acids potentially accessible for intestinal absorption. However, this method provides no information regarding the specific amino acids absorbed (essential or non-essential) and the metabolic efficiency in utilizing these amino acids, which is what ultimately influences growth. Consequently, outcomes from in vitro trials offer only partial insights into the metabolic utilisation of proteins. They primarily serve as a tool for selecting protein ingredients based on their total amino acid release under simulated digestive conditions specific to a given species.

### 4.5. Histological Analysis

Understanding the impact of dietary sources on fish pathology requires expertise in animal histology [77]. In general, animal proteins with low levels of antinutritional factors, provided their freshness and quality are satisfactory, tend not to induce liver or intestinal pathologies. Conversely, plant proteins with high levels of these factors are more likely to cause such issues [78].

In our study, feeding with organic byproducts of the Iberian pig, either with or without organic insect meal, appeared to induce liver steatosis in seabass and the fish fed the CON diet. The accumulation of lipid droplets is usually, but not always, associated with a malfunction of metabolism or related to fat metabolism. Other authors have observed steatosis in fish, possibly due to reduced energy availability [79] and liver damage associated with alternative ingredients such as ring-dried blood and feather meals [80]. However, cod fed a combination of wheat gluten and soybean protein concentrate at a maximum supplementation of 44% exhibited no histological alterations in the liver [81].

## 5. Conclusions

Diets containing organic alternative raw ingredients yield high growth parameters but result in differences from fish fed the control diet. Most parameters measuring efficiency, such as FCR, PER, PPV, and PFV, show no differences from the control diet, and some parameters, such as apparent digestibility of gross energy or the hydrolysis rates, even yield very promising values. The fish fed the TR-IN and IB-TR diets showed more favourable growth performance than fish fed the IB or IB-IN diet, although slightly lower growth performance than the CON group. This result shows that the organic ingredients used in the diets have potential as more sustainable substitutes for fishmeal. Ultimately, meals derived from byproducts may have significant appeal owing to their cost-competitiveness relative to fish meal, rendering them a potentially attractive and economical choice in organic aquaculture diets.

## Figures and Tables

**Figure 1 animals-13-03816-f001:**
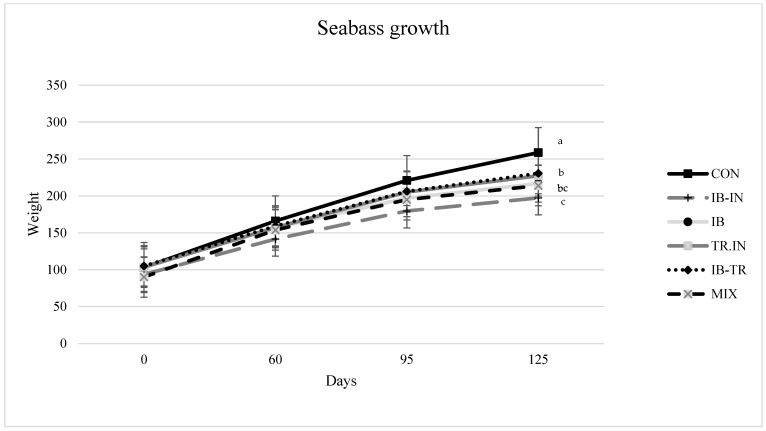
Increase in the average weight of the fish during the experiment. Values represented as means (*n* = 3). The different letters in each sampling indicate significant differences (*p* < 0.05) by the Newman-Keuls test. CON: control; IB-IN: Insect-Iberic; IB: Iberic; TR-IN: Trout-Insect; IB-TR: Trout-Iberic; and MIX: Insect-Iberic-Trout.

**Figure 2 animals-13-03816-f002:**
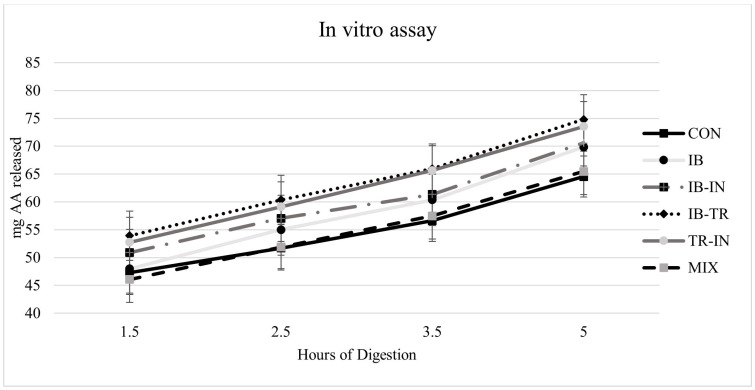
Release results of amino acids that cross the membrane after hydrolysing from the protein in all experimental diets. Values represented as means (*n* = 3). (*p* < 0.05) by the Newman-Keuls test. CON: control; IB-IN: Insect-Iberic; IB: Iberic; TR-IN: Trout-Insect; IB-TR: Trout-Iberic; and MIX: Insect-Iberic-Trout.

**Figure 3 animals-13-03816-f003:**
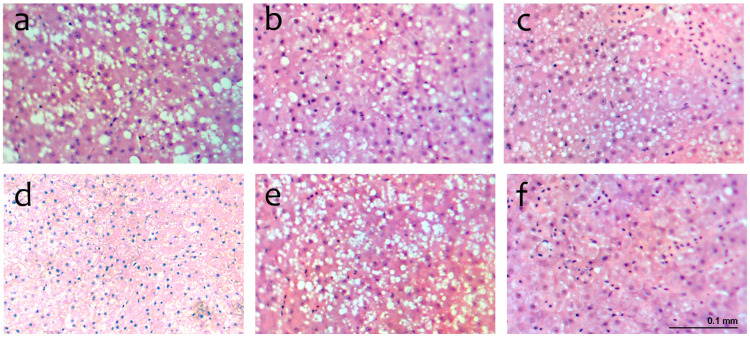
Histological details of the livers of seabass fed experimental diets (10×). (**a**) CON: control; (**b**) IB: Iberic; (**c**) IB-IN: Iberic- Insect; (**d**) IB-TR: Iberic-Trout (**e**) TR-IN: Trout-Insect; and (**f**) MIX: Insect-Iberic-Trout. Haematoxylin-Eosin staining.

**Table 1 animals-13-03816-t001:** The characteristics of the raw materials. Macronutrient composition of the different ingredients used in the study (% m.s.).

Raw Materials (%)	Fishmeal	Insect Meal	Remains of Rainbow Trout	Iberian Pork Viscera	Organic Wheat	Organic Soybean Meal
Dry matter	91.9	92.6	95.71	92.6	92.3	92.3
Crude protein	73.9	37.6	76.71	53.0	12.7	43.1
Crude lipid	11.2	28.5	17.36	28.6	1.3	9.3
Ash	14.3	13.9	11.38	3.8	1.7	6.3
Gross energy (kJ/g) **	22	24	24	27	18	21

** Gross energy (kJ/g) = [51.8 × (%C/100)) − (19.4 × (%N/100)].

**Table 2 animals-13-03816-t002:** Ingredients and proximal composition of the diets utilised in the growth experiment.

Ingredients (g kg^−1^)	CON	IB	IB-IN	IB-TR	TR-IN	MIX
Raw materials (g kg^−1^)						
Fishmeal	304					
Insect meal			214		215	143
Remains of rainbow trout				261	261	143
Iberian pork viscera		351	214	138		143
Organic wheat	213	147	75	152	80	100
Organic soybean meal	325	325	325	325	325	325
Fish oil ^a^	50	62	57	49	44	51
Organic soybean oil ^b^	83					
Calcium phosphate	10	25	25	25	25	25
Vegetable methionine ^c^	5	40	40	20	20	30
Vegetable lysine ^d^		40	40	20	20	30
Vitamins ^e^	10	10	10	10	10	10
Nutritional composition (% DM) *						
Dry matter	87.9	88.9	84.4	89.1	92.7	92.7
Crude protein	44.3	42.8	44.1	43. 7	45.1	45.6
Crude lipid	19.8	21. 6	19.7	21.0	20.0	19.8
Ash	8.3	6.8	8.9	9.4	11.1	9.4
Gross energy **	22.4	23.4	23.5	23.4	23.1	23.2
Digestible energy	77.7	83.9	76.5	83.4	73.3	78.3

^a^ Fish oil (Industrias Afines, SRL (Arpo), Polgono industrial A Veigadaa, Ra as Baloutas, de Abaixo, 24, 36416, Pontevedra, Spain). ^b^ Organic soybean oil (Clearspring Ltd., Acton Park Estate, London W3 7QE, UK). ^c^ Vegetable methionine (Adibio S.L.|Edificio Galileo, C/Enebros 74, 2^a^ planta|44002 Teruel (Spain). ^d^ Vegetable lysine (Adibio S.L.|Edificio Galileo, C/Enebros 74, 2^a^ planta|44002 Teruel (Spain). ^e^ Vitamin-and-mineral mix (g kg^−1^): Premix: 25; Choline, 10; DL-a-tocopherol, 5; ascorbic acid, 5; (PO_4_)_2_Ca_3_, 5. Premix composition: retinol acetate, 1,000,000 IU kg^−1^; calciferol, 500 IU kg^−1^; DL-a-tocopherol, 10;m menadione sodium bisulfite m menadione, 0.8; thiamine hydrochloride, 2.3; riboflavin, 2.3; pyridoxine hydrochloride, 15; cyanocobalamine, 25; nicotinamide, 15; pantothenic acid, 6; folic acid, 0.65; biotin, 0.07; ascorbic acid, 75; inositol, 15; betaine, 100; polypeptides 12. CON: control; IB-IN: Insect-Iberic; IB: Iberic; TR-IN: Trout-Insect; IB-TR: Trout-Iberic; and MIX: Insect-Iberic-Trout. * The nutritional composition values represent the means of five analyses conducted during each cycle of feed production throughout the experiment. ** Gross energy (kJ/g) = [51.8 × (%C/100)) − (19.4 × (%N/100)].

**Table 3 animals-13-03816-t003:** Concentrations of essential and nonessential amino acids in experimental diets.

Diets.	CON	IB	IB-IN	IB-TR	TR-IN	MIX
Essential amino acids (g 100 g^−1^)						
Arginine	3.01	2.27	2.25	2.07	2.42	2.67
Histidine	1.13	2.13	1.14	0.99	1.20	1.10
Isoleucine	1.71	2.04	1.83	1.69	1.96	1.85
Leucine	3.22	3.72	3.18	3.02	3.10	3.24
Lysine	2.83	2.72	2.50	2.59	2.70	2.65
Methionine	0.78	0.51	0.74	0.97	0.90	0.56
Phenylalanine	1.90	2.17	1.94	1.73	1.91	1.93
Threonine	1.65	1.76	1.47	1.35	1.29	1.64
Valine	2.29	2.71	2.44	2.15	2.30	2.44
Non-essential amino acids (g 100 g^−1^)						
Alanine	2.31	2.64	2.33	2.08	2.25	2.54
Aspartic acid	3.89	4.14	4.28	3.69	4.14	3.96
Cysteine	0.47	0.33	0.42	0.48	0.47	0.32
Glutamic acid	6.72	6.94	5.78	6.13	5.85	6.19
Glycine	2.35	3.08	2.16	2.53	2.54	2.50
Proline	2.05	2.33	2.14	2.02	2.38	2.24
Serine	1.78	3.32	2.24	1.74	2.13	1.87
Tyrosine	1.43	1.57	1.87	1.21	1.85	1.70

CON: control; IB-IN: Insect-Iberic; IB: Iberic; TR-IN: Trout-Insect; IB-TR: Trout-Iberic; and MIX: Insect-Iberic-Trout.

**Table 4 animals-13-03816-t004:** The specific conditions of the protein hydrolysis assay.

	Acid Stage	Alkaline Stage
E:S ratio (U/mg protein) *	4.0	8.5
Ph	3.5	8.5
Time (hours)	1.5	3.5
Temperature (°C)	25	25

* E:S ratio: enzyme/substrate ratio.

**Table 5 animals-13-03816-t005:** Growth and nutritional parameters of European seabass fed experimental diets.

Diets	CON	IB	IB-IN	IB-TR	TR-IN	MIX	SEM
Initial weight (g)	92.0	91.2	88.2	89.5	91.2	90	2.15
Final weight (g)	258.7 ^a^	216.7 ^bc^	197.3 ^c^	230.7 ^b^	227.3 ^b^	214.0 ^bc^	6.55
Survival rate (%)	95.0	97.3	94.0	95	96.3	97.7	2.82
SGR (% day^−1^) ^1^	1.16 ^a^	1.06 ^bc^	1.01 ^c^	1.10 ^b^	1.09 ^b^	1.05 ^bc^	0.018
FI (g100 g^−1^ fish day^−1^) ^2^	0.95	0.95	1.02	0.96	0.99	1.01	0.039
FCR ^3^	2.02	2.42	2.64	2.36	2.38	2.12	0.174
PER ^4^	0.91	0.82	0.82	0.82	0.75	0.80	0.088

Values represented as means (*n* = 3). SEM: standard error of the mean. Different superscripts in the same row indicate significant differences (*p* < 0.05) by the Newman-Keuls test. CON: control; IB-IN: Insect-Iberic; IB: Iberic; TR-IN: Trout-Insect; IB-TR: Trout-Iberic; and MIX: Insect-Iberic-Trout. ^1^ SGR = 100 × ln (final weight/initial weight)/days. ^2^ FI (g 100 g fish^−1^ day^−1^) = 100 × feed consumption (g)/average biomass (g) × days. ^3^ FCR = feed consumption (g)/biomass gain (g). ^4^ PER = biomass gain (g)/protein intake (g).

**Table 6 animals-13-03816-t006:** Body composition and retention efficiencies of seabass at the beginning of the experiment and after feeding experimental diets.

	Initial	CON	IB	IB-IN	IB-TR	TR-IN	MIX	SEM
Dry matter %	29.9	39.9	38.1	39.2	39.0	40.6	38.8	1.3
Crude protein %	18.0	16.8	16.5	16.7	16.9	17.4	16.3	0.4
Crude fat %	8.4	19.2	18.0	18.6	18.0	19.4	18.4	1.1
Ash %	3.1	3.1	3.4	4.3	3.5	3.1	3. 5	0.4
PPV ^1^ %		15.9	13.7	12.8	16.9	15.9	13.7	1.2
PFV ^2^ %		64.9	56.0	59.4	58.0	64.3	61.4	4.4
Retention efficiencies essential amino acids (%) *								
Arginine		20.3 ^ab^	14.7 ^b^	20.7 ^ab^	26.95 ^ab^	31.9 ^a^	14.8 ^b^	3.12
Histidine		14.5 ^ab^	5.5 ^b^	13.7 ^ab^	16.1 ^a^	21.2 ^a^	13.0 ^ab^	2.10
Isoleucine		17.6	11.3	11.7	18.2	17.7	12.9	1.92
Leucine		14.1	8.2	11.3	15.8	18.6	11.0	2.25
Lysine		19.9	12.9	18.7	23.7	27.2	15.7	3.29
Methionine		11.9	24.5	14.1	12.9	15.5	23.8	3.02
Phenylalanine		12.5	8.1	10.7	15.3	16.9	10.6	1.96
Threonine		17.5	12.0	15.6	20.9	28.2	14.4	2.23
Valine		14.7	9.0	11.7	17.4	18.9	11.6	2.63

Values represented as means (*n* = 3). SEM: standard error of the mean. Different superscripts in the same row indicate significant differences (*p* < 0.05) by the Newman-Keuls test. CON: control; IB-IN: Insect-Iberic; IB: Iberic; TR-IN: Trout-Insect; IB-TR: Trout-Iberic; and MIX: Insect-Iberic-Trout. ^1^ PPV% = Protein retained (final fish protein × Final biomass (g)) × 100 − Initial fish protein × initial biomass (g)/Protein ingested (kg of ingested feed ×% crude protein). ^2^ PFV% = Fat retained (final fish fat × Final biomass (g)) × 100 − Initial fish fat × initial biomass (g)/fat ingested (kg of feed × % crude fat). * Retention efficiencies of amino acids (%) = (fish amino acid gain (g) × 100)/amino acid intake (g).

**Table 7 animals-13-03816-t007:** Apparent digestibility coefficients of dry matter and nutrients of seabass fed experimental diets.

	Diets						
ADC (%) *	CON	IB	IB-IN	IB-TR	TR-IN	MIX	SEM
Dry matter	55.6 ^bc^	62.2 ^a^	53.0 ^c^	59.2 ^ab^	43.3 ^d^	53.5 ^c^	1.8
Phosphorus	52.0 ^bc^	68.3 ^a^	44.5 ^c^	57.8 ^b^	25.0 ^d^	57.0 ^b^	3.0
Crude protein	88.3 ^abc^	91.6 ^ab^	87.1 ^bc^	91.8 ^a^	85.3 ^c^	89.7 ^abc^	1.4
Gross energy	77.7 ^bc^	83.9 ^a^	76.5 ^c^	83.4 ^ab^	73.3 ^c^	78.3 ^abc^	2.0

Values represented as means (*n* = 3). SEM: standard error of the mean. Different superscripts in the same row indicate significant differences (*p* < 0.05) by the Newman-Keuls test. CON: control; IB-IN: Insect-Iberic; IB: Iberic; TR-IN: Trout-Insect; IB-TR: Trout-Iberic; and MIX: Insect-Iberic-Trout. * ADC. ADCdm = 100 − [100 × (% Cr_2_O_3_ in diet/% Cr_2_O_3_ in faeces)]. ADCnut = 100 − [100 × (% feed marker/% faeces marker) × (% nutrient. energy. amino acid. or fatty acid in urine/% of nutrient. Energy. amino acid. or fatty acid in feed)].

**Table 8 animals-13-03816-t008:** Linear adjustment equations for the protein-hydrolysis dynamics of the six experimental diets.

Diet	Linear Adjustment	Time (h)
CON	y = 4.9449x + 39.573	12.22 ^a^
IB	y = 6.1678x + 39.016	9.89 ^b^
IB-IN	y = 5.5398x + 42.66	10.35 ^bc^
MIX	y = 5539x + 37.906	11.21 ^a^
IB-TR	y = 5.9287x + 45.235	9.24 ^d^
TR-IN	y = 5.9574x + 44.145	9.37 ^cd^

y: degree of protein hydrolysis, x: time. Different superscripts in the same row indicate significant differences (*p* < 0.05) by the Newman-Keuls test. CON: control; IB-IN: Insect-Iberic; IB: Iberic; TR-IN: Trout-Insect; IB-TR: Trout-Iberic; and MIX: Insect-Iberic-Trout.

**Table 9 animals-13-03816-t009:** Histological assessments of the livers of seabass fed experimental diets.

Diets	CON	IB	IB-IN	IB-TR	TR-IN	MIX	SEM
Nucleus diameter (μm)	4.8 ^a^	3.3 ^c^	3.4 ^c^	4.1 ^b^	4.1 ^b^	3.6 ^c^	0.12
Hepatocyte diameter (μm)	11.2 ^a^	7.7 ^b^	8.2 ^b^	8.9 ^b^	8.4 ^b^	8.6 ^b^	0.24

Values represented as means (*n* = 100). SEM: standard error of the mean. Different superscripts in the same row indicate significant differences (*p* < 0.05) by the Newman-Keuls test. CON: control; IB-IN: Insect-Iberic; IB: Iberic; TR-IN: Trout-Insect; IB-TR: Trout-Iberic; and MIX: Insect-Iberic-Trout.

## Data Availability

Data are contained within the article.

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
