# Peer review of "Organic Ingredients as Alternative Protein Sources in the Diet of Juvenile Organic Seabass (Dicentrarchus labrax)"

_animals, 2023, doi:10.3390/ani13243816_

Round 1

Reviewer 1 Report

Comments and Suggestions for Authors

Line 139: please indicate N-NH4, N-NO2 ecc

Diet and feeding: in my opinion could be appreciate if you insert a table with raw materials characteristics (indicated from line176 to line 181, at least of test ingredients)

Table 1: the diet are not similar, the protein and fat level are not similar between the diet and this will be compromise the reliability of the data). It is missing the energy value (Gross and Digestible energy) for each feed. 

Table 2: also the level of aminoacid are different: methionine (essential aa that needs some addition in the feed when the fish meal is reduced) or Lysine, are very important in low fish meal diet. These low levels compromised the results.

Line 275: you have descibed ADC, but where are the results? in Table 5, you indicated the Retention efficiencies but this parameters are not ADC values!

Table 5: PPV1 and PFV2 what means 1 and 2? 

Line 602: where are the results of P?

Conclusion: in my opinion the results were not positive, these raw materials cannot substitute in this trial the fish meal, probably due to the differences between the diets, in terms of energy and aminoacids content.

Author Response

Dear Editor

We have carefully read the reviewers’ comments regarding the manuscript recently submitted to Animals," Organic Ingredients as Alternative Protein Sources in the Diet of Juvenile Organic Seabass (Dicentrarchus labrax).” Below, we respond to each of the suggestions by the reviewers. The manuscript has been thoroughly revised in response to the reviewers' recommendations. The manuscript has been reviewed by a B.A. and M.A. in English to ensure the language quality. We will re-upload a manuscript that responds to comments and changes and hope the revised manuscript will be published in ANIMALS smoothly.

Response to Reviewer:

Thank you for your feedback on our study. We appreciate the thoughtful review and constructive feedback provided by the reviewer.

Point 1: Line 139: please indicate N-NH4, N-NO2 ecc

Revised and added line 179.

Point 2: Diet and feeding: in my opinion could be appreciate if you insert a table with raw materials characteristics (indicated from line176 to line 181, at least of test ingredients)

Thank you for your suggestion; the table with test ingredient characteristics added. Table 2

Point 3: Table 1: the diet are not similar, the protein and fat level are not similar between the diet and this will be compromise the reliability of the data). It is missing the energy value (Gross and Digestible energy) for each feed. 

While the formulations were initially designed to maintain a consistent composition of 45% crude protein (CP) and 20% crude lipid (CL), challenges in effectively mixing certain ingredients led to variations in some of these values. Indicated in lines 162-165. the calculated gross energy value was also added to the table.

Point 4: Table 2: also the level of aminoacid are different: methionine (essential aa that needs some addition in the feed when the fish meal is reduced) or Lysine, are very important in low fish meal diet. These low levels compromised the results.

Thank you for your comment regarding the amino acid levels in Table 3. We acknowledge the importance of adequate levels of essential amino acids, particularly methionine and lysine, in low-fish meal diets. The challenge of organic feeding means that synthetic amino acids cannot be used. The organic amino acids available are precursors of amino acids and amino acids mix and do not allow the perfect balance of amino acids as can be done with synthetic ones. This challenge is what it means to look for 100% organic alternatives to fish meal.

Lines 588-593 have change to include now this point

Point 5: Line 275: you have descibed ADC, but where are the results? in Table 5, you indicated the Retention efficiencies but this parameters are not ADC values!

We would like to clarify that we possess the essential data, including the Apparent Digestibility Coefficients (ADC) for dietary crude protein, gross energy, phosphorus, and dry matter. These data were initially included in our work but were inadvertently omitted during the final revision before submission to the journal. We apologize for this oversight. The missing data are available and properly included in Table 7. We regret any confusion caused by their absence and appreciate the opportunity to rectify this error.

Point 6: Table 5: PPV1 and PFV2 what means 1 and 2? 

A description of these values was added below the table 6.

Point 7: Line 602: where are the results of P?

As we clarify in point 5, the results of P were added in Table 7.

Point 8: Conclusion: in my opinion, the results were not positive; these raw materials cannot substitute in this trial the fish meal, probably due to the differences between the diets in terms of energy and amino acids content.

Thank you for your insightful comments. We fully acknowledge the challenges associated with formulating organic diets for carnivorous fish, considering nutritional requirements, organic regulations, and the availability of organic feed ingredients. Our primary objective aligns with the broader goal of enhancing sustainability in carnivorous fish production by exploring viable alternatives to fishmeal (FM) that are both effective and economically feasible.

Your assessment rightly highlights the potential deficiencies in certain amino acids in the tested organic ingredients. Additionally, we acknowledge the inclusion of vegetable lysine and methionine in the diets to meet the specific requirements of sea bass. These adjustments were made to address any nutritional gaps and optimize the formulation to support the health and growth of the fish.  

Meals derived from by-products may present significant appeal owing to their cost-competitiveness relative to fish meal, rendering them a potentially attractive and economical choice.

The manuscript has been reviewed by a B.A. and M.A. in English to ensure the language quality.

Finally, we appreciate the reviewer's feedback and take these comments seriously. Your suggestions will help us enhance the quality and rigor of our research. We are committed to addressing these concerns and conducting further studies to provide more robust and comprehensive results in the future. Thank you for your valuable input.

Sincerely yours,

The authors

Reviewer 2 Report

Comments and Suggestions for Authors

This study aims to evaluate organic ingredients as alternative protein sources in the diet of juvenile Seabass. However, the experimental design is not reasonable enough, especially the level of organic ingredients added to the feed. At the same time, some data are lacking, such as ADC of dietary crude protein, lipid and dry matter, or unreliable, such as data from the PI and DI measurements in the control group.

Line 25-29 The description of the experimental diets needs to be rewritten, as it cannot fully reflect the design of the feed formulation.

Line 37-39 The conclusion section should be an objective conclusion based on experimental results. Please rewrite the conclusion section.

Line 140 Please provide the start and end times of feeding trial, as well as the daily natural light duration.

Line 153-159 Please explain the reference basis for the addition of organic ingredients to replace fish meal in the experimental diets. For example, the IB-IN diet contained the same level of insect meal (214) and Iberian pork viscera (214), while the IB-TR diet contained the different level of rest of rainbow trout (261) and Iberian pork viscera (138).

Line 246 Survival (S) should be changed to survival rate (SR).

Line 269 Why not use yttrium trioxide as an inert standard? Cr2O3 as an inert standard for determining digestibility is not appropriate because of its toxicity. Therefore, at present, Cr2O3 has rarely been used the digestibility of fish.

Line 341-342 IB-IB-TR? Which diet?

Line 363 Why is ADC of dietary crude protein, lipid and dry matter not provided?

Line 366-367 Please provide the calculation formulas for PPV, PFV and retention efficiencies essential amino acids.

Table 5: Please provide the units of the data.

Line 430-431 The PI and DI measurements in the control group seem to be unreliable. Because, even if the values of the control group are higher than those of the experimental group, it seems doubtful that it is several times higher. Because similar results were not observed in the results of the growth.

Line 577 Why does the author not provide data on the apparent digestibility of diets?

Line 579 Where is the data on protein digestibility?

Line 582-584 The amino acid composition in experimental diets does not support this hypothesis.

Line 591-593 Where is the data on protein and energy ADCs?

Line 604 Provide the data of the ADC of phosphorus.

Line 622-624 The results of in vitro digestibility was inconsistent with the growth performance. Please explain.

Line 653-655 The control diet containing fish meal induced liver steatosis in seabass. This result seems to be inconsistent with common sense.

Comments on the Quality of English Language

No.

Author Response

Dear Editor

We have carefully read the reviewers’ comments regarding the manuscript recently submitted to Animals,"Organic Ingredients as Alternative Protein Sources in the Diet of Juvenile Organic Seabass (Dicentrarchus labrax).” Below, we respond to each of the suggestions by the reviewers. The manuscript has been thoroughly revised in response to the reviewers' recommendations. The manuscript has been reviewed by a B.A. and M.A. in English to ensure the language quality. We will re-upload a manuscript that responds to comments and changes and hope the revised manuscript will be published in ANIMALS smoothly.

This study aims to evaluate organic ingredients as alternative protein sources in the diet of juvenile Seabass. However, the experimental design is not reasonable enough, especially the level of organic ingredients added to the feed. At the same time, some data are lacking, such as ADC of dietary crude protein, lipid and dry matter, or unreliable, such as data from the PI and DI measurements in the control group.

 Response to Reviewer:

Thank you for your feedback on our study. We appreciate the thoughtful review and constructive feedback provided by the reviewer. We understand the importance of a robust experimental design and the need to address potential shortcomings.

Regarding the levels of organic ingredients in the feed, it's essential to clarify that the ingredient levels were designed to ensure an equivalent protein content across all diets. The primary objective was not to maintain the same ingredient level but to achieve uniform protein content in each diet

In response to the reviewer's comment about missing data, we would like to clarify that we possess the essential data, including the Apparent Digestibility Coefficients (ADC) for dietary crude protein, gross energy, phosphorus, and dry matter. These data were initially included in our work but were inadvertently omitted during the final revision before submission to the journal. We apologize for this oversight. The missing data and biometric indices are available and included in Table 7. We regret any confusion caused by their absence and appreciate the opportunity to rectify this error.

Point 1: Line 25-29 The description of the experimental diets needs to be rewritten, as it cannot fully reflect the design of the feed formulation.

Now, The description of the experimental diets lines 24-31 is:

A total of 486 juvenile seabass with an average weight of 90 g were fed six diets containing organic protein sources: the control group (CON) was fed a diet with conventional fishmeal from sustainable fisheries as the primary protein source. The other five groups were fed diets featuring different compositions: the organic Iberian pig meal by-product (IB diet), a combination of organic Iberian pig meal by-product and insect meal (IB-IN diet), the mix of organic Iberian pig meal by-product and organic rainbow trout meal by-product (IB-TR diet), a blend of organic rainbow trout meal by-product and insect meal (TR-IN), and a mixed diet combining all these protein sources (MIX diet) as protein sources.

Point 2: Line 37-39 The conclusion section should be an objective conclusion based on experimental results. Please rewrite the conclusion section.

 The conclusion section was rewritten in lines 42-45

Point 3: Line 140: Please provide the start and end times of the feeding trial and the daily natural light duration.

 Revised and added lines 176 and 181.

Point 4: Line 153-159 Please explain the reference basis for the addition of organic ingredients to replace fish meal in the experimental diets. For example, the IB-IN diet contained the same level of insect meal (214) and Iberian pork viscera (214), while the IB-TR diet contained the different levels of rest of rainbow trout (261) and Iberian pork viscera (138).

Regarding the levels of organic ingredients in the feed, it's essential to clarify that the ingredient levels were designed to ensure an equivalent protein content across all diets. The primary objective was not to maintain the same ingredient level but to achieve uniform protein content in each diet. I will try to make it more clear now in line 203-207:

“The proximal composition of the raw materials is shown in Table 2. Formulations were initially designed with different inclusions of raw ingredients to maintain a consistent composition of 45% crude protein (CP) and 20% crude lipid (CL); challenges in effectively mixing certain ingredients led to variations in some of these values (Table 1).”

Point 5: Line 246 Survival (S) should be changed to survival rate (SR).

 Revised and added line 307.

Point 6: Line 269: Why not use yttrium trioxide as an inert standard? Cr2O3 as an inert standard for determining digestibility, is inappropriate because of its toxicity. Therefore, at present, Cr2O3 has rarely been used for the digestibility of fish.

We have been using Cr2O3 (chromium oxide) for many years as a standard to determine digestibility. But you are right; our latest studies show that the use of yttrium oxide is safer. All our studies after this one will use Yttrium. In fact, we are preparing a paper about it. 

Point 7: Line 341-342 IB-IB-TR? Which diet?

 Revised and corrected line 443. IB-TR diet

Point 8: Line 363 Why is ADC of dietary crude protein, lipid, and dry matter not provided?

 The authors would like to clarify that we possess the essential data, including the Apparent Digestibility Coefficients (ADC) for dietary crude protein, gross energy, phosphorus, and dry matter. These data were initially included in our work but were inadvertently omitted during the final revision before submission to the journal. We apologize for this oversight. The missing data are available and adequately included in Table 7. We regret any confusion caused by their absence and appreciate the opportunity to rectify this error.

Point 9: Line 366-367: Please provide the calculation formulas for essential amino acids' PPV, PFV, and retention efficiencies.

 Revised and added below the table 6 and in the main text, lines 322.

Point 10: Table 5: Please provide the units of the data.

 It was revised and added in Table 6.

Point 11: Line 430-431 The PI and DI measurements in the control group seem to be unreliable. Because, even if the values of the control group are higher than those of the experimental group, it seems doubtful that it is several times higher. Because similar results were not observed in the results of the growth.

We have thoroughly reviewed our data and regret to inform you that there are inaccuracies, rendering the data unreliable. The error occurred when a researcher inadvertently combined the samples with those from other similar trials. Consequently, we have decided to exclude all the intestine histology data from the paper. We feel very disappointed about it. We sincerely appreciate the reviewer bringing this to our attention and apologize for any inconvenience this may cause.

Point 12: Line 577 Why does the author not provide data on the apparent digestibility of diets?

As the authors clarified in point  8, data on the apparent digestibility of diets was added to Table 7.

Point 13: Line 579 Where is the data on protein digestibility?

As indicated by the authors in point 8, information regarding the apparent digestibility of diets has been incorporated into Table 7.

Point 14: Line 582-584 The amino acid composition in experimental diets does not support this hypothesis.

Thank you for the advice; the sentence has now been rewritten: “Besides, the trout and Iberian pig meals obtained a higher hydrolysis curve and better histidine retention.” Line 713-715.

Point 15: Line 591-593 Where is the data on protein and energy ADCs?

As indicated by the authors in point 8, information regarding the apparent digestibility of diets has been incorporated into Table 7.

Point 16: Line 604 Provide the data of the ADC of phosphorus.

As indicated by the authors in point 8, information regarding the apparent digestibility of diets has been incorporated into Table 7.

Point 17: Line 622-624 The results of in vitro digestibility were inconsistent with the growth performance. Please explain.

This paragraph was added to explain this to readers, lines 787-798.

The inconsistency can also be frequently found when comparing data of crude protein digestibility estimated in vivo and growth performance. In the case of in vitro hydrolysis assays, as occurs in other animals, only measure the bioaccessibility of the protein fraction of the substrate (how susceptible is this nutrient to the action of the enzymes), and this makes an estimation of the potential bioavailability (amount of amino acids potentially available for intestinal absorption. No information about the type of amino acids absorbed (essential or not) and the metabolic efficiency in the use of such amino acids is provided, finally resulting in growth. In this sense, results of the in vitro trials can be only partially informative on the metabolic use of proteins and can be used mainly as a selection tool for protein ingredients in terms of their total amino acid release under the simulated digestive conditions of a given species.

Point 18: Line 653-655 The control diet containing fish meal induced liver steatosis in seabass. This result seems to be inconsistent with common sense.

You are correct that the observation of liver steatosis in the control diet, which typically includes fish meal, may seem inconsistent with common sense. But Liver steatosis, also known as fatty liver disease, is generally associated with the accumulation of fat in the liver and is often linked to high-fat diets or metabolic imbalances; in our experience, this amount of steatosis is not so high to reflect a pathological effect, and just a lightly overfeed.

The manuscript has been reviewed by a B.A. and M.A. in English to ensure the language quality.

Finally, we appreciate the reviewer's feedback and take these comments seriously. Your suggestions will help us enhance the quality and rigor of our research. We are committed to addressing these concerns and conducting further studies to provide more robust and comprehensive results in the future. Thank you for your valuable input.

Sincerely yours,

The authors

Reviewer 3 Report

Comments and Suggestions for Authors

Comments and Suggestions for Authors

On the manuscript entitled “ Organic Ingredients as Alternative Protein Sources in the Diet of Juvenile Organic Seabass (Dicentrarchus labrax)”, with reference number animals-2709823, authors explored the inclusion of organic ingredients as protein sources in the diet of juvenile seabass. With this purpose, authors have tested 6 diets including fishmeal as the control group and other five groups with varying proportions of organic ingredients, including organic insect meal (IN), organic Iberian pig meal by-product (IB), and organic rainbow trout meal by-product (TR) as protein sources during 125-day feeding trial. The issue here covered is interesting as try to solve one of the main bottlenecks on the sustainable growth and development of the aquaculture, reducing the dependency of fish meal in aquafeeds. Although it is already known that these protein sources can be partially included in aquafeeds, the complete replacement of these sources might benefit their implementation in the aquafeeds. Authors should address several issues before the present manuscript can be accepted for publication in Animals journal.

Lines 156-157: The diet (no.5) is composed of rainbow trout by-products and organic insect meal as protein sources. It is better to remove FM in this sentence because it may be made a mistake for reader of the manuscript.

 Line 271: In this sentence “……the amount of chromium oxide in diets and urine after acid digestion…”, it can be a mistake of urine. It should be feces because the marker (chromic oxide) is used in the diet, so it is seen in the feces.

Line 277: In the equation 1, the “ feces” should be corrected. It is written as faeces.

Line 298: What has been the aim of authors for measuring amylase?

Line 296: As the fish have the real stomach and this organ is separated from the fish, it seems that the authors have assessed Pepsin. Thus, acid protease is not true and it is better to correct to pepsin.

Line 305: In table 3, the authors have shown the conditions used in the protein hydrolysis assay. The main question in this case can be the kind of protease used for hydrolysis in vitro system. The authors have shown the ration of E:S, but what is the kind of protease?

Line 366: The authors have shown the results of the fat efficiency retention (PFV) while in the materials and methods section and subsection of “Indices of Growth and Biometric Parameters” is not observed anything about this factor.

 Line 415: In table 7, the liver of trout.., why trout? This work is conducted in seabass.

Comments on the Quality of English Language

The quality of English language is good but it can be improved.

Author Response

Dear Editor

We have carefully read the reviewers’ comments regarding the manuscript recently submitted to Animals," Organic Ingredients as Alternative Protein Sources in the Diet of Juvenile Organic Seabass (Dicentrarchus labrax).” Below, we respond to each of the suggestions by the reviewers. The manuscript has been thoroughly revised in response to the reviewers' suggestions. The manuscript has been reviewed by a B.A. and M.A. in English to ensure the language quality. We will re-upload a manuscript that responds to comments and changes and hope the revised manuscript will be published in ANIMALS smoothly.

Authors should address several issues before the present manuscript can be accepted for publication in Animals Journal.

Response to Reviewer:

Thank you for your feedback on our study. We appreciate the thoughtful review and constructive feedback provided by the reviewer.

Point 1: Lines 156-157: The diet (no.5) is composed of rainbow trout by-products and organic insect meal as protein sources. It is better to remove FM in this sentence because it may be made a mistake for the reader of the manuscript.

Removed and corrected line 199.

Point 2: Line 271: In this sentence “……the amount of chromium oxide in diets and urine after acid digestion…”, it can be a mistake of urine. It should be feces because the marker (chromic oxide) is used in the diet, so it is seen in the feces.

Revised and corrected line 348.

Point 3: Line 277: In the equation 1, the “ feces” should be corrected. It is written as faeces.

Revised and corrected throughout the manuscript.

Point 4: Line 298: What has been the aim of the authors for measuring amylase?

Sorry, this was a mistake. The authors measured amylase activity initially as a part of the preliminary evaluation of the enzyme equipment of the juvenile European seabass. Nevertheless, the effect of such an enzyme in the breakdown of carbohydrate substrates used for the in vitro trials was not evaluated in the present study. The sentence has been modified accordingly. Line 380.

Point 5: Line 296: As the fish have the real stomach and this organ is separated from the fish, it seems the authors have assessed Pepsin. Thus, acid protease is not true, and it is better to correct pepsin.

As indicated in the same paragraph, semipurified extracts obtained from the digestives of juvenile sea bass were used to run the in vitro assays. A new paragraph has been written for clarification: Revised and corrected line 380.

Point 6: Line 305: In table 3, the authors have shown the conditions used in the protein hydrolysis assay. The main question in this case can be the kind of protease used for hydrolysis in vitro system. The authors have shown the ration of E:S, but what is the kind of protease?

As indicated in the same paragraph, semipurified extracts obtained from the digestives of juvenile sea bass were used to run the in vitro assays. A new paragraph has been written for clarification:  lines 380/393

The extracts were prepared by mechanical homogenization of the tissues in distilled water (1:10 w/v) followed by centrifugation (3,220 . g, 20 min, 4°C). The supernatant was then filtered through a dialysis system with a MWCO of 10 kDa (Pellicon XL, Millipore), and the concentrated extracts were freeze-dried until required for the assays. Activities of acid protease in the stomach (pepsin) and total intestinal alkaline proteases present in the extracts were measured using the methods indicated in refs 48 and 49. The values of protease activities were used as indicators to estimate the amount of extracts required to provide physiological enzyme-substrate ratios in the assays. These were calculated considering, on the one hand, the average total production of enzyme measured in several fish in relation to their live weight and, on the other, the average intake per meal of fish of such size obtained from commercial ration tables.

Point 7: Line 366: The authors have shown the results of the fat efficiency retention (PFV) while in the materials and methods section and subsection of “Indices of Growth and Biometric Parameters” is not observed anything about this factor.

Revised and added protein and fat efficiency retention factor in a subsection of “Indices of Growth, line 309.

Point 8: Line 415: In table 7, … the liver of trout.., why trout? This work is conducted in seabass.

 Revised and corrected line 539. Was a mistake.

Comments on the Quality of English Language

The quality of the English language is good, but it can be improved.

The manuscript has been reviewed by a B.A. and M.A. in English to ensure the language quality.

We appreciate the reviewer's feedback and take these comments seriously. Your suggestions will help us enhance the quality and rigor of our research. We are committed to addressing these concerns and conducting further studies to provide more robust and comprehensive results in the future. Thank you for your valuable input.

Sincerely yours,

The authors

Reviewer 4 Report

Comments and Suggestions for Authors

Review for the paper “Organic Ingredients as Alternative Protein Sources in the Diet of Juvenile Organic Seabass (Dicentrarchus labrax)” by Eslam Tefal, Ignacio Jauralde, Silvia Martínez-Llorens, Ana Tomás-Vidal, María Consolación Milián-Sorribes, Francisco Javier Moyano, David S. Peñaranda, Miguel Jover-Cerdá submitted to "Animals".

Aquaculture currently accounts for approximately 50% of human fish consumption. However, the future progress of aquaculture may face significant challenges due to the escalating prices of fish oil and fishmeal. To overcome these obstacles, scientific research and feed manufacturers are diligently working towards developing alternative protein sources for fish diets. This is crucial to ensure proper nutrition for animal growth while reducing reliance on traditional protein sources.

In recent years, notable progress has been made in exploring alternative protein sources for aquafeeds. Plant-based ingredients, as well as fisheries and aquaculture by-products, have shown great potential in providing the necessary protein for aquafeeds over the next 10 to 20 years. Additionally, the utilization of insect meals as a protein source has also been investigated. To investigate the impact of different diets on juvenile seabass, the authors conducted a laboratory study. The diets used in the study varied in their protein sources, and were evaluated based on their effects on growth performance, feed utilization, survival rates, digestibility, and histological parameters. The authors specifically examined the use of insect meal, pork meal by-products, and organic pork meal by-products as potential protein sources. The results of the study indicated that while some diets showed promising results, they were unable to fully substitute the control diet. The performance indices of these alternative protein sources were lower compared to the control diet. Despite these limitations, the findings of this study provide a valuable foundation for future research in the field of aquaculture. The identification of potential alternative protein sources and their effects on seabass aquaculture has significant implications for the industry. Further research can build upon these findings to develop optimized diets that strike a balance between nutritional requirements and economic feasibility in the aquaculture sector.

Recommendations:

L 330. It is crucial to assess whether the data used in the study meet the assumptions of normal distribution and homogeneity of variances for the parametric approach used by the authors. It would be informative to indicate whether the authors tested their data for these assumptions and provide details about the specific tests used.

L 332. There is a discrepancy in the text regarding the representation of error. While the authors state that "The findings are represented as mean±standard error," they refer to "standard deviation" in the text (L 344). It would be beneficial for the authors to clarify which type of error (standard error or standard deviation) was used and revise the text accordingly to ensure consistency and accuracy.

Figure 1. The y-axis label in Figure 1 should be corrected to "Weight, g" rather than "Growth," as weight is the intended variable being represented.

Also, it is pertinent to address the concern regarding the sample size (n = 3) used for statistical comparisons. This sample size may be considered too small to obtain reliable results. However, it is worth mentioning that a larger sample size (n = 100) was used for histological comparisons, which is deemed appropriate.

L 344. The authors stated that "Different superscript in each sampling means significant differences (p < 0.05)," but no superscript letters are visible in Figure 1. The authors should rectify this discrepancy and ensure that the appropriate superscript letters are included in the figure to indicate significant differences.

Table 4. It is unclear for which parameter the standard error of the mean (SEM) was calculated in Table 4. The authors should present the error (SEM) for each value in the table, specifying which parameter it corresponds to, to provide clarity and enhance the interpretation of the results.

L 358-362, 379. The information mentioned in these lines should be moved to the "Materials and Methods" section, as they are more relevant to the methodology rather than the main text.

L 390. The inclusion of information about the statistical test, specifically mentioning "Test Newman-Keuls," in the caption of a figure (Figure 5 in this case) requires an explanation. The authors should provide a rationale for including this information in the caption to enhance the readers' understanding.

L 395. The authors made a statement about the "steeper slope of the IB diet's adjustment line" suggesting faster hydrolysis. To support this conclusion, it would be beneficial for the authors to apply an appropriate statistical analysis, such as ANCOVA, to examine the significance of the observed difference and reinforce the validity of their claim.

Furthermore, it is advisable for the authors to review the literature, as some important details seem to be missing in various sections (L 756, 821, 835, 851, 894, 896, 907).

Specific remarks.

L 24. Consider replacing “weight of 90 g fish” with “weight of 90 g”

L 65. Consider replacing “74.032 tonnes” with “74,032 tonnes”

L 65. Consider replacing “The show is up 60%” with “This marks a 60% increase”

L 66. Consider replacing “46.341” with “46,341”

L 78. Consider replacing “There needs to be more research” with “More research is required”

L 215. Consider replacing “Three species” with “Three specimens”

L 273. Consider replacing “All analyzes” with “All analyses”

L 331. Consider replacing “variance analysis with a Newman-Keul test for comparison of means” with “analysis of variance with a Newman-Keul test for multiple comparisons”

L 429. Consider replacing “differences have been observed” with “differences were observed”

L 577. Consider replacing “and see how” with “and examine how”

L 625. Consider replacing “were compatible” with “are consistent”

L 657. Consider replacing “[81]. Or liver damage” with “[81] as well as liver damage”

L 686. Consider replacing “About the mucosa alteration, our findings” with “Our findings”

L 801. Consider replacing “Reviews in Aquaculture. Mar” with “Reviews in Aquaculture.”

Comments on the Quality of English Language

Minor.

Author Response

Dear Editor

We have carefully read the reviewers’ comments regarding the manuscript recently submitted to Animals," Organic Ingredients as Alternative Protein Sources in the Diet of Juvenile Organic Seabass (Dicentrarchus labrax).” Below, we respond to each of the suggestions by the reviewers. The manuscript has been thoroughly revised in response to the reviewers' suggestions. The manuscript has been reviewed by a B.A. and M.A. in English to ensure the language quality. We will re-upload a manuscript that responds to comments and changes and hope the revised manuscript will be published in ANIMALS smoothly.

Comments and Suggestions for Authors

Thank you for your feedback on our study. We appreciate the thoughtful review and constructive feedback provided by the reviewer.

Recommendations:

Point 1: L 330. It is crucial to assess whether the data used in the study meet the assumptions of normal distribution and homogeneity of variances for the parametric approach used by the authors. It would be informative to indicate whether the authors tested their data for these assumptions and provide details about the specific tests used.

Revised and added:  All data were checked for normal distribution and homogeneity of variances. line 428.  Using the Statgraphics® Plus 5.1 statistical program (Statistical Graphics Corp., Rockville, MO, USA),

Point 2: L 332. There is a discrepancy in the text regarding the representation of error. While the authors state that "The findings are represented as mean±standard error," they refer to "standard deviation" in the text (L 344). It would be beneficial for the authors to clarify which type of error (standard error or standard deviation) was used and revise the text accordingly to ensure consistency and accuracy.

Thank you for your comment. The authors have consistently used the standard error of the mean (SEM) throughout their data analysis and presentation. Revised throughout the manuscript and corrected was a mistake.

Point 3: Figure 1. The y-axis label in Figure 1 should be corrected to "Weight, g" rather than "Growth," as weight is the intended variable being represented.

 Revised and corrected.

Point 4: Also, it is pertinent to address the concern regarding the sample size (n = 3) used for statistical comparisons. This sample size may be considered too small to obtain reliable results. However, it is worth mentioning that a larger sample size (n = 100) was used for histological comparisons, which is deemed appropriate.

We appreciate the reviewer's comments and would like to provide further clarification regarding the sample size and its rationale in our study. In our experimental design, the sample size of n = 3 represented the number of tanks for each treatment, and each tank served as an experimental unit. The experiment was conducted in triplicate for each treatment, resulting in a total of 18 tanks, which corresponds to the number of tanks in our laboratory setup. While we acknowledge that a sample size of n = 3 for statistical comparisons may be considered, in general, relatively small. But is generally accepted because each tank represents several animal means   (27 in our study), and feed is taken as one data per tank. Due to practical constraints, such as tank availability and resource limitations, we opted to conduct the experiment with triplicate tanks for each treatment. This sample size was determined based on the available resources and the feasibility of our experimental setup.

Conversely, a larger sample size of n = 100 was chosen for histological comparisons to comprehensively assess histological parameters. This decision was made to ensure we accurately captured potential variations in the histological analysis. In this case, n=100 is possible cause several measures can be taken of each tank opposite to feed intake.

While we recognize that larger sample sizes for statistical comparisons are generally preferred, the use of triplicate tanks, as described above, was a practical choice for our specific experiment and widely accepted in aquaculture. We believe that the combination of statistical analyses with the larger histological sample size offers a balanced approach to addressing the research questions of our study.

Point 5: L 344. The authors stated that "Different superscript in each sampling means significant differences (p < 0.05)," but no superscript letters are visible in Figure 1. The authors should rectify this discrepancy and ensure that the appropriate superscript letters are included in the figure to indicate significant differences.

Revised and added the appropriate superscript letters are included in the figure to indicate significant differences.

Point 6: Table 4. It is unclear for which parameter the standard error of the mean (SEM) was calculated in Table 4. The authors should present the error (SEM) for each value in the table, specifying which parameter it corresponds to, to provide clarity and enhance the interpretation of the results.

We appreciate the reviewer's feedback and comments. Table 4 displays the standard error of the mean (SEM), denoted as "Standard Error (grouped s)," which serves as a measure of the variability in our sampling. The SEM is calculated by dividing the pooled standard deviation by the number of observations at each level.

Point 7: L 358-362, 379. The information mentioned in these lines should be moved to the "Materials and Methods" section, as they are more relevant to the methodology rather than the main text.

  Revised and corrected and moved to "Materials and Methods" section as main text and also as foot.  Was as table foot.

Point 8: L 390. The inclusion of information about the statistical test, specifically mentioning "Test Newman-Keuls," in the caption of a figure (Figure 5 in this case) requires an explanation. The authors should provide a rationale for including this information in the caption to enhance the readers' understanding.

The authors have reviewed Figure 5 and its caption but cannot locate the specific case in question. If there is an error in this information, it may require further clarification or correction.

Point 9: L 395. The authors made a statement about the "steeper slope of the IB diet's adjustment line" suggesting faster hydrolysis. To support this conclusion, it would be beneficial for the authors to apply an appropriate statistical analysis, such as ANCOVA, to examine the significance of the observed difference and reinforce the validity of their claim.

Thanks to the reviewer for the suggestion. A comparison of slopes has been carried out with the specific module of Statgraphics Centurion XV. The statistical significance is now included in Table 8.

Point 10: Furthermore, it is advisable for the authors to review the literature, as some important details seem to be missing in various sections (L 756, 821, 835, 851, 894, 896, 907).

We appreciate the reviewer's comments and would like to clarify that the authors have reviewed the literature, and it appears to be accurate without any missing information. The citations have been verified on Google Scholar.

Specific remarks.

Point 11: L 24. Consider replacing “weight of 90 g fish” with “weight of 90 g”

 Revised and corrected Line 25.

Point 12: L 65. Consider replacing “74.032 tonnes” with “74,032 tonnes”

 Revised and corrected Line 78.

Point 13: L 65. Consider replacing “The show is up 60%” with “This marks a 60% increase”

 Revised and corrected Line 79.

Point 14: L 66. Consider replacing “46.341” with “46,341”

 Revised and corrected Line 78.

Point 15: L 78. Consider replacing “There needs to be more research” with “More research is required”

 Revised and corrected Line 95.

Point 16: L 215. Consider replacing “Three species” with “Three specimens”

  Revised and corrected Line 276.

Point 17: L 273. Consider replacing “All analyzes” with “All analyses”

 Revised and corrected Line 351.

Point 18: L 331. Consider replacing “variance analysis with a Newman-Keul test for comparison of means” with “analysis of variance with a Newman-Keul test for multiple comparisons”

  Revised and corrected Line 432.

Point 19: L 429. Consider replacing “differences have been observed” with “differences were observed”

 Revised and corrected Line 467.

Point 20: L 577. Consider replacing “and see how” with “and examine how”

 Revised and corrected Line 708.

Point 21: L 625. Consider replacing “were compatible” with “are consistent”

 Revised and corrected Line 760.

Point 22: L 657. Consider replacing “[81]. Or liver damage” with “[81] as well as liver damage”

 Revised and corrected Line 811.

Point 23: L 686. Consider replacing “About the mucosa alteration, our findings” with “Our findings”

 Revised and this frase are deleted from the paper.

Point 24: L 801. Consider replacing “Reviews in Aquaculture. Mar” with “Reviews in Aquaculture.”

Revised and corrected.

Comments on the Quality of English Language

Minor.

The manuscript has been reviewed by a B.A. and M.A. in English to ensure the language quality.

We appreciate the reviewer's feedback and take these comments seriously. Your suggestions will help us enhance the quality and rigor of our research. We are committed to addressing these concerns and conducting further studies to provide more robust and comprehensive results in the future. Thank you for your valuable input.

Sincerely yours,

The authors

Round 2

Reviewer 1 Report

Comments and Suggestions for Authors

Please invert Table 1 and 2, before you can explain the caracteristic of raw material and after indicate the formulation of the diets

Table 1 and 2: insert the DE value

ADC: In my opinion is not possible compare the digestibility trial and the growth results because the trials were conducted in two steps; in my opinion the adc data could be removed in to the article.

Author Response

Dear Editor

We have carefully read the reviewer 1 comments regarding the manuscript recently submitted to Animals," Organic Ingredients as Alternative Protein Sources in the Diet of Juvenile Organic Seabass (Dicentrarchus labrax).” Below, we respond to each of the suggestions by the reviewer 1. The manuscript has been thoroughly revised in response to the reviewer 1 recommendations. We will re-upload a manuscript that responds to comments and changes and hopes the revised manuscript will be published in ANIMALS smoothly.

Response to Reviewer:

Thank you for your feedback on our study. We appreciate the thoughtful review and constructive feedback provided by the reviewer.

Point 1: Please invert Table 1 and 2, before you can explain the caracteristic of raw material and after indicate the formulation of the diets.

We appreciate the reviewer's suggestion. Revised and inverted tables 1 and 2.

Point 2: Table 1 and 2: insert the DE value

We appreciate the reviewer's suggestion. In response, we have included the digestible energy (DE) values for the diets in Table 2, providing comprehensive information on the energy content of each formulated diet. However, it is important to note that the DE values for individual ingredients have not been included in this study. The calculation and inclusion of DE values for individual ingredients were not performed due to, (time limitations or other factors].

We hope the addition of DE values for the diets in Table 2 enhances the overall clarity and utility of the presented data.

Point 3: ADC: In my opinion is not possible to compare the digestibility trial and the growth results because the trials were conducted in two steps; in my opinion, the adc data could be removed in to the article.

We appreciate the reviewer's careful consideration of our work and the suggestion to remove ADC data from the article. We would like to provide clarification on the separation of digestibility trials and growth results.

The decision to conduct digestibility trials and growth studies in separate tanks was deliberate and based on the specific requirements of each type of trial. Digestibility trials necessitate controlled conditions with specific tank setups to accurately measure nutrient absorption. But overall, growth tanks are not prepared to collect faeces.  In contrast, growth studies demand different parameters, and combining both in a single set of tanks could introduce confounding variables.

The fish used to study digestibility come from the same budget as the fish for growth trials.  We acknowledge the challenges in directly comparing these two sets of data; when a direct measure is totally necessary, the most used technique is probably the stripping technique, but it has been pointed out that it alters digestion to a certain extent and harms the animal. We believe that presenting both aspects contributes valuable insights to understanding the overall impact of the dietary interventions.

We appreciate the opportunity to address this concern and are thankful for the thoughtful evaluation of our work.

Finally, we appreciate the reviewer's feedback and take these comments seriously. Your suggestions will help us enhance the quality and rigor of our research. We are committed to addressing these concerns and conducting further studies to provide more robust and comprehensive results in the future. Thank you for your valuable input.

Sincerely yours,

The authors

Reviewer 2 Report

Comments and Suggestions for Authors

No.

Comments on the Quality of English Language

No.

Author Response

Thank you for your feedback on our study. We appreciate the thoughtful review and constructive feedback provided by the reviewer.

no comments to respond. 

Reviewer 3 Report

Comments and Suggestions for Authors

The manuscript can be accepted in present form.

Author Response

Thank you for your feedback on our study. We appreciate the thoughtful review and constructive feedback provided by the reviewer.

There are no comments to respond to.